# Molecular detection of SARS-CoV-2 using a reagent-free approach

**Ronan Calvez\*, Andrew Taylor, Leonides Calvo-Bado, Donald Fraser, Colin G. Fink** 

Micropathology Limited, University of Warwick Science Park, Coventry, United Kingdom

\* r.calvez@micropathology.com

## Abstract

Shortage of reagents and consumables required for the extraction and molecular detection of SARS-CoV-2 RNA in respiratory samples has led many laboratories to investigate alternative approaches for sample preparation. Many groups recently presented results using heat processing method of respiratory samples prior to RT-qPCR as an economical method enabling an extremely fast streamlining of the processes at virtually no cost. Here, we present our results using this method and highlight some major pitfalls that diagnostics laboratories should be aware of before proceeding with this methodology. We first investigated various treatments using different temperatures, incubation times and sample volumes to optimise the heat treatment conditions. Although the initial data confirmed results published elsewhere, further investigations revealed unexpected inhibitory properties of some commonly used universal transport media (UTMs) on some commercially available RT-qPCR mixes, leading to a risk of reporting false-negative results. This emphasises the critical importance of a thorough validation process to determine the most suitable reagents to use depending on the sample types to be tested. In conclusion, a heat processing method is effective with very consistent Ct values and a sensitivity of 96.2% when compared to a conventional RNA extraction method. It is also critical to include an internal control to check each sample for potential inhibition.

## Introduction

A novel severe acute respiratory syndrome coronavirus-2 (SARS-CoV-2) emerged in the Wuhan province of China in December 2019 and is now recognised as the cause of the present human coronavirus pandemic (Covid-19) [1–3]. As of 12th May 2020, 4,088,848 cases have been confirmed globally with 283,153 deaths [3]. Whole genome sequencing data from infected patients suggests a very recent shift into humans with the closest related coronaviruses found in bats and pangolins [4]. There are now more than several thousand publicly available whole genome sequences for SARS-CoV-2 which will allow for extensive studies on virus adaptation and aid vaccine development. A recent study analysed 7,666 SARS-CoV-2 genome sequences and provided evidence of possible adaptation to its human host [5]. The main symptoms of SARS-CoV-2 infection are fever, dry cough, dyspnea, headache and pneumonia [6]. These symptoms can progress to respiratory failure and death and this has been observed

**Competing interests:** The authors have declared
that no competing interests exist.

more frequently in elderly people, or those with underlying health conditions and also may
have an ethnic association. Loss of sense of smell and/or taste also appear to be symptoms asso-
ciated with SARS-CoV-2 infection. The disease is primarily transmitted by human-to-human
contact although some studies have suggested that the virus will survive on surfaces for up to 3
days [7]. Due to the unprecedented high demand for SARS-CoV-2 testing required to identify
and isolate infected individuals, many laboratories are now facing shortages in reagents and
consumables required for nucleic acid (NA) extraction. This led us to an investigation of alter-
native sample processing methods that would remove the requirements for NA extraction and
allow rapid diagnosis [8–11]. Examining viral RNA in respiratory samples directly would aid
high-throughput screening and sample preparation at low cost. In March 2020, Fomsgaard
and Rosenstierne [9] reported results of experiments using heat processing of respiratory sam-
ples prior to RT-qPCR as a very attractive method enabling an extremely fast streamlining of
the diagnostic processes. Here, we provide some recommendations based on our results and
highlight some major pitfalls that diagnostics laboratories should be aware of when processing
patient samples for molecular detection of SARS-CoV-2.

## Materials and methods

### Patient samples

All patient samples used in this study were received at Micropathology Ltd for SARS-CoV-2
testing from various NHS Trusts in the UK. These included dry swabs, swabs resuspended at
source in various transport media and nasopharyngeal aspirates (NPA). Dry swabs were re-
suspended upon receipt in 500 μL 0.1% Igepal CA-630 (Sigma-Aldrich) using sterile disposable
Pasteur pipettes. For all other swabs and NPA, the original resuspension buffer received was
used directly. All samples were initially tested using our reference SARS-CoV-2 assay with
prior NA extraction (NA/ABI protocol, see below). Selected SARS-Cov-2-positive (including
samples with a very low viral load as measured by ddPCR) and SARS-Cov-2-negative respira-
tory samples were re-tested following heat treatment. Samples were chosen to cover the most
common swab sample types received at Micropathology Ltd for SARS-Cov-2 testing, and for
each swab types, viral loads ranging from less than 1,000 (Ct>37.0) to greater than 100,000
(Ct<30.0) SARS-Cov-2 RNA copies/mL for each swab type.

### Heat treatment protocol

We investigated various treatments using different temperatures, incubation times and sample
volumes based on data published by Fomsgaard *et al.* in March 2020 [9] using heat processing
methodology of respiratory samples prior to RT-qPCR for the detection of SARS-CoV-2. To
verify this methodology, 25 μL to 40 μL volumes of respiratory samples were transferred to a
96 well PCR plate in a microbiological safety category II cabinet. Plates were the sealed and
heat treated at either 95˚C or 98˚C for 2 to 20 minutes on a fully calibrated PCR block. At the
end of the incubation, the samples were transferred to another thermal block set at 4˚C for at
least 5 minutes. Twenty μL of each lysate was then transferred to a 96-well PCR plate contain-
ing various commercially available 1-step RT-qPCR mixes for SARS-CoV-2 detection, apart
from the Quantabio and Promega RT-PCR mixes, where 5 μL of template was used (Table 3).
The assay was then performed and the Ct values were recorded for each sample.

### SARS-CoV-2 detection and PCR internal controls

Primer and probe (TIBMOLBIO, Berlin, Germany) sequences and concentrations are summa-
rised in S1 Table. All PCR mixes used were prepared according to the manufacturer's

recommendations and are summarised in S2 Table. Detection was performed on Roche Light-Cycler® 480 (Roche Diagnostics) instruments.

Our reference in-house SARS-Cov-2 assay included NA extraction using a modified Maxwell® HT 96 gDNA kit (Promega Corp) with automated extraction on the KingFisher FLEX platform (Thermo Fisher Scientific Inc). SARS-CoV-2 detection (2019-nCoV_N1 assay derived from the CDC.gov website [12]) was then performed on 20 μL extracted NA (50 μL reaction) using the ABI TaqMan® Fast Virus 1-Step RT-qPCR kit (#4444436, Thermo Fisher Scientific Inc). The cycling conditions consisted of a 5 minutes incubation at 50˚C (reverse transcription), 20 seconds at 95˚C (RT inactivation and denaturation) followed by 45 amplification cycles (3 seconds at 95˚C and 30 seconds at 55˚C). For simplicity, this method is abbreviated NA/ABI in the text and figures below. The lower limit of detection of this assay was established at 364 SARS-CoV-2 RNA copies/mL using droplet digital PCR (QX200, Bio-Rad) for quantification. Since all the samples tested were swab samples, Ct values, rather than quantitative values, were provided as this sample type does not allow a consistent and reliable viral load estimation (variability in resuspension volume and inconsistency in surface area swabbed). For clinical purposes, samples with a Ct value greater than 39.0 were considered "weakly positive", whereas samples with a Ct value less than 20.0 were considered "strongly positive". Samples with a Ct ranging from 20.1 to 38.9 were reported as "positive".

The performance of various commercially available RT-PCR buffers was evaluated on SARS-CoV-2 detection sensitivity from heat-treated samples. Each buffer was used according to manufacturer's instructions. The Meridian Fast 1-Step RT-PCR mix produced the best results in the widest range of sample types and was therefore selected for most of the experiments performed on heat-treated samples. The cycling conditions using this mix consisted of a 10 minutes incubation at 45˚C (reverse transcription), 2 minutes at 95˚C (RT inactivation and denaturation) followed by 45 amplification cycles (5 seconds at 95˚C and 20 seconds at 55˚C).

The baculovirus *Adoxophyes orana* granulovirus (AoGV) is an insect DNA virus routinely used at Micropathology Ltd as an extraction and PCR internal control. In our routine diagnostics, the whole virus is spiked at a fixed concentration in the lysis buffer prior to sample extraction and detected using an in-house TaqMan hydrolysis probe-based assay targeting the granulin gene of AoGV. In the present study, 4 μL of heat-treated samples or molecular-grade water were added to 6 μL AoGV RT-PCR reaction mixture (10 μL final volume) containing extracted AoGV DNA ($2\times10^5$ genomes/mL) and the primers and probes for AoGV (S1 Table). PCR inhibition was monitored in this stand-alone reaction by comparing AoGV levels in heat treated samples to its levels in non-heat treated samples (water template). AoGV amplification and detection was performed using the Meridian Fast 1-Step RT-PCR using the cycling conditions described above.

Potato Virus Y (PVY) RNA was used as an RT-PCR internal control to monitor the presence of potential RT and PCR inhibitors. Briefly, a synthetic RNA encoding part of the PVY coat protein gene was synthesised (Ultramer® RNA Oligo, IDT). PVY RNA was then spiked at $1\times10^7$ copies/mL in the SARS-CoV-2 PCR reaction (prepared using the Meridian Fast 1-Step RT-PCR mix, see above) together with the primers and probe for PVY (S1 Table). Simultaneous detection of SARS-CoV-2 and PVY was performed using the cycling conditions described above. Due to the difficulty to source the synthetic RNA, the PVY internal control could not be tested on the samples used in the initial stages of this study since the samples used were no longer available.

## Statistical analysis

Statistical Analysis was carried out in R v3.6.2 [13].

### Ethical statement

This study was performed at Micropathology Ltd (University of Warwick Science Park, Coventry, UK) in April 2020. Patient samples were de-identified and were not considered Human Subjects Research due to the quality improvement and public health intent of the work. The performance of this study was reviewed and approved by the Micropathology Ltd Ethics Committee Review Board composed of Professor Sheila Crispin (MA, VetMB, DVA, DVOphthal, DipEVCO, FRCVS), Professor Christopher Dowson (BSc, PhD), Rt Hon Countess of Mar, Most Rev Dr Gordon Mursell (MA, Hon DD) and William NH Taylor (BTech)). No additional consent was necessary.

## Results and discussion

### Optimisation of the heat treatment method and inhibitory properties of Universal Transport Medium (UTM)

To evaluate the best assay conditions, we initially tested a random range of SARS-CoV-2-positive and negative patient samples (n = 16) and carried out heat treatment at different temperatures and different incubation times. SARS-CoV-2 detection was then performed using the ABI TaqMan® Fast Virus 1-Step RT-qPCR kit. All the dry swab (resuspended in 0.1% Igepal at Micropathology Ltd) and green-cap swab (MWE, re-suspended in Σ-Virocult® transport medium at source) samples tested were correctly detected using a 5 minutes incubation at 95°C followed by a 4°C-incubation for at least 5 minutes (Table 1). These conditions yielded a comparable sensitivity to the one obtained using our reference assay (with consistent Ct values across both methods), supporting the previously published data using a similar method [5]. Shorter (<2 minutes) and longer (10 minutes and 20 minutes) incubation times at these temperatures, or at 90°C, did not improve sensitivity (Table 1). Surprisingly, all the red-cap tubes for swab samples tested (COPAN swabs re-suspended in UTM-RT medium, Copan Diagnostics) failed to produce any detectable signal despite a significant viral load (Ct<37.0). Original results reported in our laboratory were obtained using our conventional NA/ABI protocol (described in the methods section) also using the ABI TaqMan® Fast Virus 1-Step RT-qPCR kit.

### Performance of a Meridian Bioscience RT-qPCR mixes

In order to understand the failure of the heat treatment prior RT-qPCR detection of SARS-CoV-2 in the samples re-suspended in UTM, we investigated an additional RT-qPCR kit (Fast 1-Step RT-qPCR from Meridian Biosciencce) using another range of SARS-CoV-2-positive patient samples (n = 22) previously detected using our reference assay (NA/ABI). These samples included a range of viral loads for each of the most common swab sample types received at Micropathology Ltd for SARS-CoV-2 testing. The two strongest samples tested in Table 1 (001529 and 001502) were also included in this new cohort of samples. Both RT-PCR kits (ABI TaqMan® Fast Virus 1-Step RT-qPCR, and Meridian Fast 1-Step RT-qPCR) were tested following a 5 minutes heat treatment at 95°C in 40 μL volume and used according to the manufacturer's instructions. Supporting our findings (Table 1), the ABI TaqMan® One-Step RT-qPCR mix systematically failed to detect SARS-CoV-2 in all swab samples re-suspended in UTM (COPAN swabs) even in the presence of viral loads greater than $1 \times 10^6$ copies/mL (003850 and 003862) (Table 2). In contrast, the Meridian RT-PCR mix allowed the reliable detection of SARS-Cov-2, even at very low viral loads (Ct>37.0), in all swab samples tested. This suggested two possible explanations; the presence of RT and/or PCR inhibitors, or the degradation of the RNA during the heating process in samples re-suspended in UTM. Interestingly, a range of non-swab respiratory NPA samples (3 positive and 2 negative), all failed to allow the successful

**Table 1. Optimisation of heat-induced sample preparation for SARS-CoV-2 testing targeting the N-gene and using the TaqMan® Fast Virus 1-Step RT-qPCR kit from ABI.**

| | Temperature (°C) | | 95 | | 98 | | 90 | | | | 95 | | | | 98 |
|---|---|---|---|---|---|---|---|---|---|---|---|---|---|---|---|
| | Time (min) | | 5 | | 5 | | 5 | | 10 | | 2 | 5 | 10 | 20 | 5 |
| | Reaction volume (µL) | | 25 | 50 | 25 | 50 | 25 | 50 | 25 | 50 | 30 | | | | |
| Patient num | NA/ABI[1] | Swab[2] | | | | | | | | | | | | | |
| 001508 | 35.8 | Green | 35.8 | 35.8 | 36.5 | 35.9 | 36.7 | 36.5 | 34.8 | 35.1 | 36.1 | 36.0 | 35.9 | 36.5 | 37.0 |
| 001524 | ND | Dry | ND | ND | ND | ND | ND | ND | ND | ND | ND | ND | ND | ND | ND |
| 001525 | ND | Dry | ND | ND | ND | ND | ND | ND | ND | ND | ND | ND | ND | ND | ND |
| 001528 | ND | Dry | ND | ND | ND | ND | ND | ND | ND | ND | ND | ND | ND | ND | ND |
| 001491 | 41.8 | Dry | ND | 40.1 | ND | ND | ND | ND | ND | ND | Not tested | | | | |
| 001498 | 38.2 | Dry | 36.1 | 35.4 | 37 | 37.9 | 35.8 | 36.2 | 36.4 | 35.8 | 35.4 | 34.5 | 35.4 | 35.9 | ND |
| 001518 | 37.0 | Dry | 36.8 | 35.9 | 35.0 | 36.8 | 35.6 | 34.6 | 35.4 | 35.2 | 33.6 | 34.0 | 34.4 | 35.3 | ND |
| 001512 | 35.9 | Dry | 34.0 | 35.6 | 35.5 | 35.8 | 35.8 | 35.5 | 34.5 | 34.7 | 34.6 | 34.1 | 33.8 | 35.3 | 34.9 |
| 001523 | 29.9 | Dry | 32.5 | 30.7 | 33.9 | 32.2 | 33.5 | 34.6 | 32.7 | 32.6 | 31.9 | 32.6 | 30.6 | 34.2 | 31.7 |
| 001496 | 41.8 | Red | ND | ND | ND | ND | ND | ND | ND | ND | ND | ND | ND | ND | ND |
| 001549 | 41.1 | Red | ND | ND | ND | ND | ND | ND | ND | ND | ND | ND | ND | ND | ND |
| 001499 | 40.0 | Red | ND | ND | ND | ND | ND | ND | ND | ND | ND | ND | ND | ND | ND |
| 001513 | 38.9 | Red | ND | ND | ND | ND | ND | ND | ND | ND | ND | ND | ND | ND | ND |
| 001526 | 38.9 | Red | ND | ND | ND | ND | ND | ND | ND | ND | ND | ND | ND | ND | ND |
| 001502 | 36.2 | Red | ND | ND | ND | ND | ND | ND | ND | ND | ND | ND | ND | ND | ND |
| 001529 | 35.6 | Red | ND | ND | ND | ND | ND | ND | ND | ND | ND | ND | ND | ND | ND |

[1] Results obtained following nucleic acid extraction (Custom Promega Maxwell® HT DNA kit) and purification on a KingFisher platform (ThermoFisher) and are expressed as a Ct value. Amplification performed using the ABI TaqMan® Fast Virus 1-Step RT-qPCR mix.

[2] Each swab type contained a different buffer. Green swab: Σ-Virocult® transport medium (MWE). Dry swab: resuspended in Igepal (0.1%) at Micropathology Ltd. Red Swab: universal transport medium, UTM (COPAN swab in skirted tube).

ND: Not Detected.

amplification of SARS-CoV-2 even in the presence of a viral load greater than 100,000 copies/mL (Ct<30.0). All NPA samples were prepared at source using a saline solution.

## Evidence for PCR inhibitors in respiratory samples

To determine whether the failure to detect SARS-CoV-2 in the NPA samples was caused by PCR inhibitors, an aliquot of each heat-treated sample was mixed with extracted DNA from a baculovirus (AoGV) suspension. AoGV was then amplified using our in-house assay (see Materials and Methods). As shown in S1 Fig, amplification was successful in 2 out of 5 of the NPA samples tested (006222 and 006223), with delayed amplification (13 cycle delay) in one of them (006221) and no amplification at all in the other two samples (006224 and 006225). Since the same saline solution was used for all 5 NPA samples and since SARS-CoV-2 amplification failed in at least one sample despite normal AoGV amplification, these results strongly suggested the presence of PCR inhibitors and RT inhibitors in some NPA samples (mucosal cells, bacteria) leading to the generation of false negative results. The AoGV assay, however, did not reveal the presence of PCR inhibitors in any of the UTM samples tested, suggesting that the ABI RT-qPCR kit was more sensitive than the Meridian kit to inhibitory compounds present in the UTM buffer only. Although the exact composition of each RT-PCR mix was not known, we hypothesised that the MMLV RT used in the ABI kit might be more sensitive to inhibitory compounds present in the UTM buffer than the MMLV RT used in the Meridian Fast kit. An

**Table 2. Assessment of transport medium on SARS-CoV-2 detection using two RT-qPCR mixes following a 5min heat treatment at 95˚C and 5min at +4˚C.**

| Patient num | Sample type | NA/ABI[1] | Heat Treatment | |
| --- | --- | --- | --- | --- |
| | | | ABI TaqMan | Meridian Fast |
| 004124 | Red (skirted) | 38.8 | ND | ND |
| 001502 | Red (skirted) | 36.2 | ND | 38.1 |
| 001529 | Red (skirted) | 35.6 | ND | 37.9 |
| 004147 | Red (skirted) | 34.7 | ND | 36.9 |
| 003862 | Red (skirted) | 24.0 | ND | 25.7 |
| 003850 | Red (skirted) | 23.9 | ND | 27.4 |
| 003782 | Red (conical) | 38.1 | 40.4 | ND* |
| 003776 | Red (conical) | 27.5 | 27.2 | 28.9 |
| 004152 | Green | 38.8 | 40.2 | 38.0 |
| 001508 | Green | 35.8 | 34.4 | 38.1 |
| 003691 | Green | 35.5 | 37.9 | 37.1 |
| 003785 | Dry | 36.2 | 35.7 | 37.4 |
| 001512 | Dry | 35.9 | 35.4 | 36.2 |
| 003756 | Dry | 35.0 | 34.1 | 37.7 |
| 003789 | Dry | 28.4 | 27.9 | 31.2 |
| 003796 | Dry | 27.3 | 25.9 | 29.0 |
| 003760 | Dry | 21.3 | 26.8 | 28.5 |
| 006221 | NPA | ND | ND | ND |
| 006222 | NPA | ND | ND | ND |
| 006224 | NPA | 39.7 | ND | ND |
| 006223 | NPA | 33.3 | ND | ND |
| 006225 | NPA | 30.8 | ND | ND |

[1] The original results were obtained following nucleic acid extraction (Custom Promega Maxwell® HT DNA kit), purification on a KingFisher platform (ThermoFisher) and detection using the ABI TaqMan® Fast Virus 1-Step RT-qPCR kit (NA/ABI). Results are expressed as a Ct value.

Red (skirted) swab: COPAN swab with UTM-RT for viruses, Chlamydia, Mycoplasma and Ureaplasma. Red (conical) swab: REMEL M4RT swabs with VTM for transport of viruses and Chlamydia. Green swab: Σ-Virocult swab for transport of viruses. Dry: dry swabs resuspended in 0.1% Igepal at Micropathology Ltd. NPA: nasopharyngeal aspirate prepared at source using saline buffer. ND: Not Detected.

*: SARS-CoV-2-positive (Ct 39.6) upon re-test.

RNA internal control targeting a plant virus (Potato Virus Y, PVY) was later developed (see materials and methods). Although the samples analysed above were no longer available, a cohort of 265 patient swab samples were parallel tested using either the AoGV assay or the PVY assay (S3 Table). Although the PVY assay correctly detected the three samples (1.13%) that failed to amplify AoGV (PCR inhibitors), the RNA control revealed an additional 2.26% samples (6/265) displaying either full (4/265) or partial (2/265) PVY-inhibition with normal AoGV amplification, *i.e.* RT-only inhibition. While such a high proportion of RT-specific inhibitors was not expected, these results underline the importance of the internal control choice to avoid the reporting of false negative results. Samples displaying a delayed or absence of PVY amplification were not associated with any specific buffer type suggesting the sample, rather than the buffer, as the source of the inhibition in these instances.

## Parallel assessment of other commercially available RT-qPCR kits

We selected a cohort of 67 patient samples, which was representative of all the swab types received at Micropathology Ltd and contained weak to intermediate levels of SARS-CoV-2

**Table 3. Evaluation of the sensitivity of different commercially available RT-PCR kits in different swab sample types re-tested following heat treatment (5min at 95°C) compared to nucleic acid extraction.**

| | NA/ABI | Heat Treatment | | | | |
|---|---|---|---|---|---|---|
| | | ABI TaqMan[1] | Meridian Fast 1-Step[1] | Meridian Low LOD[1] | Quantabio ToughMix[2] | Promega GoTaq[2] |
| Remel swab (M4-RT VTM) | 7/7 | 7/7 | 6/7 | 7/7 | 4/7 | 7/7 |
| MWE swab (Σ-Virocult) | 3/3 | 3/3 | 3/3 | 3/3 | 3/3 | 3/3 |
| Dry swab (Igepal) | 13/13 | 10/13 | 10/13 | 10/13 | 2/13 | 7/13 |
| Cepheid Xpert® swab (VTM) | 4/4 | 0/4 | 4/4 | 4/4 | 4/4 | 4/4 |
| Copan swab (UTM-RT) | 26/26 | 0/26 | 14/26 | 11/26 | 4/26 | 7/26 |
| Unknown[3] | 6/6 | 0/6 | 6/6 | 6/6 | 3/6 | 4/6 |
| Overall detection rate | 100.0% | 33.9% | 72.9% | 69.5% | 33.9% | 55.9% |

[1] 20μL template volume.

[2] 5μL template volume.

[3] some swab samples were received in unbranded tubes containing unknown transport media.

For each swab type, a range of SARS-CoV-2-positive samples ranging from Ct 20 to 40 were selected. Each sample was tested using the reference assay (NA/ABI) to determine the 100% base line. The same samples were then heat-treated and SARS-CoV-2 detection performed using various commercially available RT-PCR kits. For each swab type, the detection rate is indicated as number of positives samples detected / total samples tested. ABI TaqMan® Fast Virus 1-Step (#4444436), Meridian Bioscience Fast 1-Step (#MDX032), Meridian Bioscience Low LOD 1-Step (#MDX025), Quantabio qScript XLT ToughMix® (#95132–100) and Promega GoTaq® 1-Step (#A6020) RT-qPCR mixes. Full data summarised in this table can be found in S4 Table. ND: Not Detected.

(median Ct 36.6 ± 4.81 cycles, assessed by our in-house assay). After recovery from storage (-20°C) and thawing, the samples were re-tested using our conventional assay with NA extraction (NA/ABI) to assess for potential RNA degradation during storage. A proportion (8/67) of the samples that initially tested positive for SARS-CoV-2 on fresh samples were not detected following a freeze/thaw cycle (S4 Table). All of these non-confirmed samples contained a sub limit of detection viral load (median Ct = 38.9 ± 0.99 cycles), suggesting a detrimental effect of the freeze-thawing cycle on RNA recovery, especially at low RNA level.

Using the confirmed positive samples (n = 59), five commercially available RT-PCR kits were evaluated in parallel (Table 3, and S4 Table for full data). Apart from the two Meridian Bioscience kits, all the RT-PCR kits tested failed to efficiently detect SARS-CoV-2 in samples re-suspended in the UTM-RT medium used in the COPAN swabs (Table 3), suggesting that these kits were more sensitive to inhibitory components present in this buffer. The lower sensitivity of the ToughMix and GoTaq kits could also be partially attributed to the reduced amount of template used for these kits; 5 μL, as per manufacturer's recommendation. The ABI Taq-Man® Fast Virus kit also failed to allow the detection of SARS-CoV-2 in samples re-suspended in the VTM medium used in the Cepheid Xpert® swabs, whereas the Meridian Fast kit showed the best overall performance in the range of sample types tested. Taken together, these results indicate that various transport media may contain specific RT and/or PCR inhibitors that may specifically interfere with the performance of some particular RT-PCR kits. This emphasises the critical importance of a rigorous assay validation by each laboratory before proceeding with a heat treatment protocol, in order to select the most adapted RT-PCR kit depending on the samples to be tested.

## Sensitivity of the heat treatment method

To assess the sensitivity of the procedure, the data summarised in Table 3 for the reference assay (NA/ABI) and the heat treatment method (HT/Meridian Fast) were further analysed. SARS-CoV-2 detection performed directly from heat treated patient samples allowed the

successful detection of 72.9% of the samples that tested positive in the NA/ABI assay following re-extraction (S4 Table). All 16 heat-treated samples where SARS-CoV-2 amplification was unsuccessful contained a very low viral load of SARS-CoV-2 (median Ct 40.1 ± 0.95 cycles), whereas all the intermediate/strong positive samples (Ct<37.0) were correctly detected. The sensitivity of the heat treatment process calculated for these selected samples (72.9%) did not reflect the true sensitivity of the method since the strongly positive SARS-CoV-2-samples (Ct<26.0) were excluded from this cohort of patient samples and only the samples that were initially detected in the reference assay were considered in the analysis.

Multiple parallel diagnostics runs using non-screened samples (n = 545) were then performed to refine the detection rate of the heat treatment method against the reference assay. After removal of the samples showing signs of PCR inhibition using our AoGV internal control, an actual sensitivity of 96.2% was calculated, with 65 samples being positive in both assays and 457 being negative in both assays (S5 Table). Samples only positive in the NA/ABI assay (13/545) displayed slightly earlier Ct values than samples only positive in the HT/Meridian Fast assay (10/545) (Ct range 35.1–40.3 and 38.3–43.0, respectively).

## Correlation between heat treatment and NA extraction protocols

Recent reports using a similar approach (70˚C for 10 minutes) suggested a very significant increase in the Ct values following heat treatment [11]. To determine the effect of our heat treatment protocol on patient samples results, the Ct values of all the positive samples recorded in S4 and S5 Tables were compiled and analysed (S6 Table). There was a good correlation between the two methods ($R^2$ = 0.78) and, as shown in Fig 1 and S6 Table, the Ct values in the heat-treated protocol (median Ct 37.1 ± 4.94) were significantly later than the ones measured using the reference assay (median Ct 35.8 ± 4.81) (Paired t-test, t = -4.0787, df = 100, p = 0.00009). However, in our case, these results were much more similar to nucleic acid extraction methods than was previously reported [11] (average ΔCt 0.92 ± 2.36 cycles as opposed to 6.1 cycles ± 1.60 cycles). Similarly, the positive samples missed in the heat treatment method (40.7 ± 1.52) were also 0.90 cycles later than the ones missed by our reference assay (39.8 ± 1.56) (S6 Table).

## Conclusions

Heat treatment of respiratory samples prior to RT-qPCR showed an attractive methodology compared to a conventional nucleic acid extraction method. Our results verify the reproducibility and the technical feasibility of a "reagent-free" extraction process, which resulted in a detection rate of 96.2% when compared to our reference NA extraction protocol. In addition, we also found a strong correlation between the two methods based on the Ct values. The success of SARS-CoV-2 diagnostics using this methodology, however, appeared strongly dependent on the properties and robustness of the RT-PCR mix used, since some virus transport media (*e.g.* UTM from COPAN swabs) appeared incompatible with subsequent SARS-CoV-2 detection in some RT-PCR mixes, whereas some non-swab samples (*e.g.* NPA), showed PCR inhibitory properties rendering them incompatible with all RT-PCR mixes evaluated in this study. Identifying and characterising the specific inhibitory compounds present in each transport medium and linking them to susceptible components present in each RT-PCR kit would undoubtedly be of scientific interest, but was beyond the scope of this study. We recommend that a thorough study of all sample types tested should be performed prior to performing direct RT-PCR detection from heat treated samples. Although heat treatment method offers several undeniable advantages in terms of time and cost savings, our results advise great caution when using such a procedure. To avoid false-negative results, it is critical to combine the use of

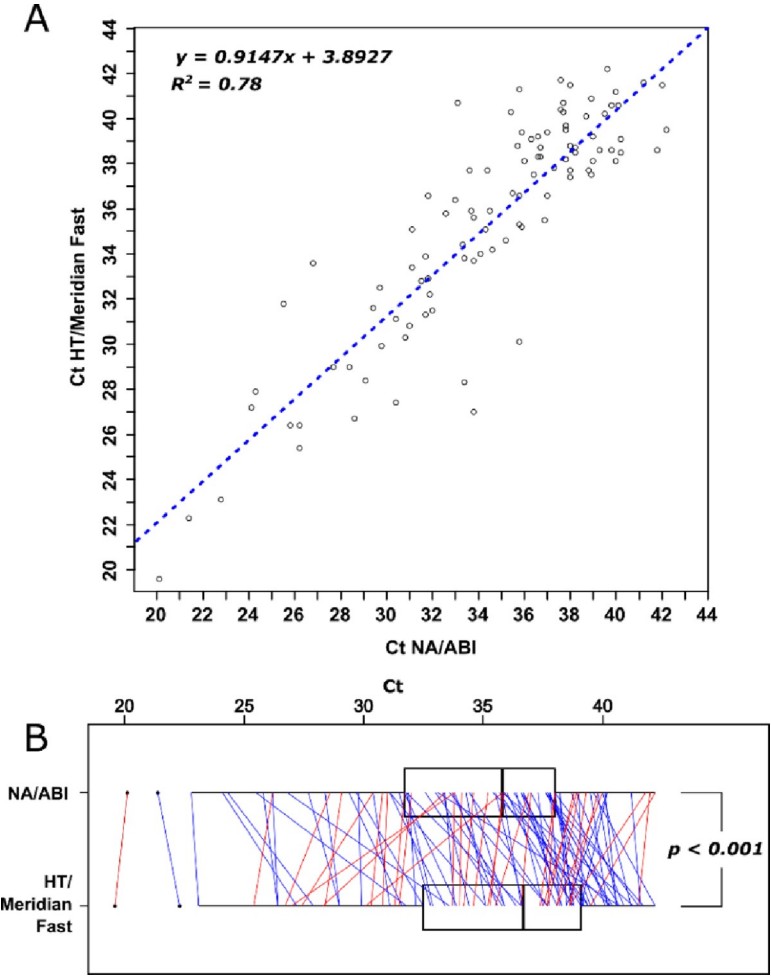

**Fig 1. Correlation between NA extraction-based and heat treatment-based detection of SARS-CoV-2.** A) Correlation of Ct values from NA/RT-qPCR (NA/ABI) and heat treatment/RT-qPCR (HT/Meridian Fast 1-Step). The Ct values obtained for the samples positive for SARS-CoV-2 using both methods (S6 Table) were plotted against each other. B) Comparison of Ct values in paired samples that were positive in both NA/ABI and Heat-treatment/RT-qPCR. The paired samples are individually plotted as red and blue lines if Heat-treatment/RT-qPCR Ct was earlier or later than NA/RT-qPCR, respectively. Paired t-test, t = -4.0787, df = 100, p = 0.00009.

internal RT and PCR controls with an alternative conventional nucleic acid procedure for samples that fail such internal controls.

## Supporting information

**S1 Table. Sequences of primers and probes for SARS-CoV-2 (N1), AoGV and PVY and optimised final concentrations for their use in real-time RT-PCR.**
(DOCX)

**S2 Table. RT-PCR mix preparation.**
(DOCX)

**S3 Table. Detection of the AoGV and PVY internal controls in multiple parallel diagnostics runs using non-screened samples (n = 265).**
(DOCX)

**S4 Table. Evaluation of the sensitivity of the heat treatment method using different commercially available RT-PCR kits.**
(DOCX)

**S5 Table. Sensitivity assessment of the heat treatment protocol.**
(DOCX)

**S6 Table. Comparison of Ct values in paired samples positive in both NA/RT-qPCR and Heat treatment/RT-qPCR.**
(DOCX)

**S1 Fig. Evidence for PCR inhibition in NPA samples.**
(DOCX)

## Acknowledgments

The authors gratefully acknowledge Dr Mark Atkins for critical review of the manuscript, the staff at Micropathology Ltd especially Dr Oliver Smith, Dr Edward Sumner and Dr Paul Scott for their scientific inputs, as well as collaborators at the University of Warwick, especially Prof. David Roper and Dr John Walsh for their support and scientific contributions.

## Author Contributions

**Conceptualization:** Ronan Calvez.

**Data curation:** Ronan Calvez.

**Formal analysis:** Ronan Calvez, Donald Fraser.

**Investigation:** Ronan Calvez.

**Methodology:** Ronan Calvez, Andrew Taylor, Leonides Calvo-Bado.

**Writing – original draft:** Ronan Calvez.

**Writing – review & editing:** Ronan Calvez, Andrew Taylor, Leonides Calvo-Bado, Donald Fraser, Colin G. Fink.

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
