## [Editor Report · Decision Letter 0]

29 Jun 2020

PONE-D-20-15666

Molecular detection of SARS-CoV-2 using a reagent-free approach

PLOS ONE

Dear Dr. Calvez

Thank you for submitting your manuscript to PLOS ONE. After careful consideration, we feel that it has merit but does not fully meet PLOS ONE’s publication criteria as it currently stands. Therefore, we invite you to submit a revised version of the manuscript that addresses the points raised during the review process.

In spite of my efforts to find reviewers for your manuscript (and many others i am in charge of reporting new data on the Cov 2 Coronavirus) reporting a simplified method for monitoringf by RTqPCR the presence of COV2 in people, i must admit that it turned out to be extremely difficult. Therefore i have no other choice than to personaly evaluate your manuscript.

Overall the manuscript reporting the monitoring of COV2 RNA by RTqPCR is simple and clear.

The major critiques are as follows:

1- Detection of Cov2 RNA by RTqPCR does not mean that the virus is infectious (introduction section ie ref 7) . As for other reports, it seems to me that the monitoring of infectious virus should be provided in parallel.

2- Using such RTqPCR monitoring, one should be able to provide the true viral loads (as it stands for HIV infection), that most probably differ from one infected person to another one. Although these data are important from a medical perspective, they are not provided as a whole (l 169)

3- the buffer conditions (see lines 134-1554) should be available so as to optimize detection and to understand why several kits failed to detect COV2 RNA (table 2)(table 3).

4- RT inhibitors can be easily detected using recombinant RT enzyme in in vitro assays

We look forward to receiving your revised manuscript.

Kind regards,

Jean-Luc EPH Darlix, MG, Ph.D.

Academic Editor

PLOS ONE

Journal Requirements:

2. Please provide additional details regarding participant consent. In the ethics statement in the Methods and online submission information, please if the need for consent was specifically waived by your ethics committee.

3. In your Discussion, please discuss Fomsgaard et al 2020 and how your work adds to their findings.

4. Please include your tables as part of your main manuscript and remove the individual files. Please note that supplementary tables (should remain/ be uploaded) as separate "supporting information" files
---

## [Author Response · Author response to Decision Letter 0]

14 Jul 2020

Rebuttal letter

We thank the academic editor for his comments on our manuscript.

Please find below our responses to each point raised by the academic editor. We hope that we satisfyingly addressed all of them, and that the manuscript will be now suitable for publication.

Sincerely,

On behalf of all authors,

Ronan Calvez.

Academic editor:

Overall the manuscript reporting the monitoring of COV2 RNA by RTqPCR is simple and clear.

The major critiques are as follows:

1. Detection of Cov2 RNA by RTqPCR does not mean that the virus is infectious (introduction section ie ref 7) . As for other reports, it seems to me that the monitoring of infectious virus should be provided in parallel.

The aim of the methodology described in this paper was primarily to perform a rapid diagnostic testing for the detection of SARS-CoV-2 RNA (less than 24 h turnaround time), particularly in scenarios like hospitals with patients and staff. The paper did not intend to discriminate between infectious and non-infectious virus. 

2. Using such RTqPCR monitoring, one should be able to provide the true viral loads (as it stands for HIV infection), that most probably differ from one infected person to another one. Although these data are important from a medical perspective, they are not provided as a whole (l 169)

The SARS-CoV-2 detection methodology described here aimed to primarily discriminate qualitatively between the presence or absence of SARS-CoV-2 RNA in patient samples. This assay has been calibrated by ddPCR to assess the low limit of detection and quantitative values could indeed be provided if the sample was quantifiable in nature (eg whole blood, plasma…). However, all the samples tested using our methodology were swab samples which are not quantifiable in nature since the buffer volume used for resuspension was variable. In addition, for an accurate result to be given, all swab samples should be taken in the exact same way (same surface area swabbed) which is unrealistic given that the samples used in this study originated from over 200 NHS Trusts in the UK.

To answer your very relevant comment regarding the clinical relevance of the amount of virus detected, each SARS-CoV-2-positive sample was reported back to the requesting authority with a caveat stating, “strongly positive” (Ct<20) or “weakly positive” (Ct>39) according to the Ct value recorded.

Two sentences have been added l 194-200 to clarify this point:

“Since all the samples tested using this methodology were swab samples, Ct values, rather than quantitative values, were provided as this sample type does not allow a consistent and reliable viral load estimation (variability in resuspension volume and inconsistency in surface area swabbed). For clinical purposes, samples containing a Ct value greater than 39 were considered “weakly positive”, whereas samples with a Ct value less than 20 were considered “strongly positive”.

3. the buffer conditions (see lines 134-1554) should be available so as to optimize detection and to understand why several kits failed to detect COV2 RNA (table 2)(table 3).

All buffers were used according to manufacturer’s instructions. The components of those buffers however were not available from the suppliers. We did request this information to understand the nature of the failed reactions but they were not able to provide it since they were protected by intellectual property or patents.

4. RT inhibitors can be easily detected using recombinant RT enzyme in in vitro assays

We have also investigated another RNA virus internal control derived from potato virus Y (PVY) to assess the presence of RT-specific PCR inhibitors. This assay is currently being validated as an internal control consisting of PVY RNA spiked in the RT-PCR mix, which will enable us to verify the presence of RT-inhibitors within the sample. A sentence has been added to clarify this point (l 168-169):

“To assess the presence of RT-only inhibitors, another internal control derived from the potato virus Y (PVY) was used in parallel.”

---

## [Decision Letter · Decision Letter 1]

2 Sep 2020

PONE-D-20-15666R1

Molecular detection of SARS-CoV-2 using a reagent-free approach

PLOS ONE

Dear Dr. Calvez,

Thank you for submitting your manuscript to PLOS ONE. After careful consideration, we feel that it has merit but does not fully meet PLOS ONE’s publication criteria as it currently stands. Therefore, we invite you to submit a revised version of the manuscript that addresses the points raised during the review process.

Notably provide  a valuable general protocol for the rapid detection by RT-PCR of the COV2 virus as well as many other viral infections (Flu viruses, Parainfluenzae viruses, RSV) (see ref's comments)

We look forward to receiving your revised manuscript.

Kind regards,

Jean-Luc EPH Darlix, MG, Ph.D.

Academic Editor

PLOS ONE

Reviewers' comments:

Reviewer's Responses to Questions

**Comments to the Author**

1. If the authors have adequately addressed your comments raised in a previous round of review and you feel that this manuscript is now acceptable for publication, you may indicate that here to bypass the “Comments to the Author” section, enter your conflict of interest statement in the “Confidential to Editor” section, and submit your "Accept" recommendation.

Reviewer #1: (No Response)

2. Is the manuscript technically sound, and do the data support the conclusions?

Reviewer #1: Partly

3. Has the statistical analysis been performed appropriately and rigorously? 

Reviewer #1: No

4. Have the authors made all data underlying the findings in their manuscript fully available?

Reviewer #1: Yes

5. Is the manuscript presented in an intelligible fashion and written in standard English?

Reviewer #1: Yes

6. Review Comments to the Author

Reviewer #1: I share the Editor’s opinion. This work is valuable owing the crucial need to have reliable and fast detection assays of SARS-Cov2 in respiratory samples. The revised version adequately addresses most of the Editor’s questions. However, I think that the authors have everything in hands to build a ready to use detailed protocol that could contribute to standardization of quick testing of clinical respiratory (and possibly other types after validation) samples for their contents in RNA virus including SARS-CoV2. The assay would include the systematic addition of a known RNA probe (potato virus Y (PVY) ) to detect any RT and PCR inhibitors in heated samples. Note that the use of a DNA probe (baculovirus AoGV) maybe dispensable unless one wants to know which of the RT or DNA pol is inhibited. In this case, the author should provide how these two probes can be obtained, how to store them and if they are of any biological threat. The author can go to doi: 10.1371/journal.pone.0172358. eCollection 2017 as an example of how to write such protocol.

Should the authors choose not to value their work as a standard protocol, they should at least provide all the information enabling reproduction of their data including:

- Primers and PCR programme to quantified the AoGV DNA

- Primers and RT-PCR programme to quantify PVY RNA

- Results obtained on PVY probe detection after addition in UTM to determine why the ABI RT-qPCR kit does not work for RNA samples while it works with the AoGV DNA probe.

Minor:

- the reference list needs upgrading: for example, ref 7 is now published in The New England Journal of Medicine doi: 10.1056/NEJMc2004973

- numbers in tables (both in main text and supplement) should be limited to only 3 significant figures (i.e. write 33.9 instead of the (statistically) meaningless 33.91)

- Table 1: line “patient num” the cases should be left empty (or fused with the upper ones) and not filled with “ND” since the line is solely used as headings.

- Table S3: the mean Ct missed should be completed with the corresponding SD value. Mean, SD, and range values of each group would gained to be also mentioned in the text since pairs of data obtained with heating prep and in house NA purification protocol look statistically not significant (statistical analysis should be performed)

- Line 197: The Ct range “positive” (20<ct<39) added="" aside="" be="" should="" the=""></ct<39)>

7. PLOS authors have the option to publish the peer review history of their article (what does this mean?). If published, this will include your full peer review and any attached files.

Reviewer #1: No

---

## [Author Response · Author response to Decision Letter 1]

17 Nov 2020

All comments and suggestions made by Reviewer#1 have been answered. Details of which can be found in the attached Response to Reviewers document attached.

---

## [Editor Report · Decision Letter 2]

19 Nov 2020

Molecular detection of SARS-CoV-2 using a reagent-free approach

PONE-D-20-15666R2

Dear Dr. Calvez,

We’re  and personaly  I am pleased to inform you that your manuscript has been judged scientifically suitable for publication and will be formally accepted for publication once it meets all outstanding technical requirements.

Kindest regards and best wishes

Jean-Luc EPH Darlix, MG, Ph.D.

Academic Editor

PLOS ONE
---

## [Editor Report · Acceptance letter]

23 Nov 2020

PONE-D-20-15666R2 

Molecular detection of SARS-CoV-2 using a reagent-free approach 

Dear Dr. Calvez:

I'm pleased to inform you that your manuscript has been deemed suitable for publication in PLOS ONE. Congratulations! Your manuscript is now with our production department. 

Kind regards, 

on behalf of

Professor Jean-Luc EPH Darlix 

Academic Editor

PLOS ONE